# Effect of Precipitation Variation on Soil Respiration in Rain-Fed Winter Wheat Systems on the Loess Plateau, China

**DOI:** 10.3390/ijerph19116915

**Published:** 2022-06-05

**Authors:** Houkun Chu, Hong Ni, Jingyong Ma, Yuying Shen

**Affiliations:** 1State Key Laboratory of Grassland Agro-Ecosystem, Lanzhou University, Lanzhou 730020, China; chuhk19@lzu.edu.cn (H.C.); nih18@lzu.edu.cn (H.N.); majy@lzu.edu.cn (J.M.); 2College of Pastoral Agricultural Science and Technology, Lanzhou University, Lanzhou 730020, China; 3National Field Scientific Observation and Research Station of Grassland Agro-Ecosystems, Qingyang 745004, China

**Keywords:** soil respiration, precipitation variation, hysteresis, Q_10_, decouple

## Abstract

Global climate change has aggravated the hydrological cycle by changing both the amount and distribution of precipitation, and this is especially notable in the semiarid Loess Plateau. How these precipitation variations have affected soil carbon (C) emission by the agroecosystems is still unclear. Here, to evaluate the effects of precipitation variation on soil respiration (R_s_), a field experiment (from 2019 to 2020) was conducted with 3 levels of manipulation, including ambient precipitation (CK), 30% decreased precipitation (P_−30_), and 30% increased precipitation (P_+30_) in rain-fed winter wheat (*Triticum aestivum* L.) agroecosystems on the Loess Plateau, China. The results showed that the average R_s_ in P_−30_ treatment was significantly higher than those in the CK and P_+30_ treatments (*p* < 0.05), and the cumulative CO_2_ emissions were 406.37, 372.58 and 383.59 g C m^−2^, respectively. Seasonal responses of R_s_ to the soil volumetric moisture content (VWC) were affected by the different precipitation treatments. R_s_ was quadratically correlated with the VWC in the CK and P_+30_ treatments, and the threshold of the optimal VWC for R_s_ was approximately 16.06–17.07%. However, R_s_ was a piecewise linear function of the VWC in the P_−30_ treatment. The synergism of soil temperature (T_s_) and VWC can better explain the variation in soil respiration in the CK and P_−30_ treatments. However, an increase in precipitation led to the decoupling of the R_s_ responses to T_s_. The temperature sensitivity of respiration (Q_10_) varied with precipitation variation. Q_10_ was positive correlated with seasonal T_s_ in the CK and P_+30_ treatments, but exhibited a negative polynomial correlation with seasonal T_s_ in the P_−30_ treatment. R_s_ also exhibited diurnal clockwise hysteresis loops with T_s_ in the three precipitation treatments, and the seasonal dynamics of the diurnal lag time were significantly negatively correlated with the VWC. Our study highlighted that understanding the synergistic and decoupled responses of R_s_ and Q_10_ to T_s_ and VWC and the threshold of the change in response to the VWC under precipitation variation scenarios can benefit the prediction of future C balances in agroecosystems in semiarid regions under climate change.

## 1. Introduction

Global climate change has aggravated the hydrological cycle, causing changes in both the amount and distribution of precipitation [1,2]. Projections from climate models show that the probability of extreme precipitation will continue to increase in the future due to the aggravation of global warming [3], which will seriously affect the process of the global carbon (C) cycle [4]. This is particularly true for arid and semiarid regions, where terrestrial C sinks are especially sensitive to precipitation variation [5]. As a vital process in the global C cycle in terrestrial ecosystems, soil respiration (R_s_) is a critical contributor to the transport of terrestrial C to the atmosphere, and small changes in R_s_ have great impacts on the global C cycle [6]. Thus, exploring how R_s_ responds to precipitation variation will enhance our predictions of the global C cycle in the context of climate change.

Soil respiration (R_s_) is the second largest terrestrial C flux, accounting for 60–90% of total ecosystem respiration [7,8]. R_s_ is composed of autotrophic respiration (R_a_) and heterotrophic respiration (R_h_) [4]. Variations in R_s_ are attributed to changes in biotic and abiotic factors, especially soil temperature (T_s_) and soil volumetric water content (VWC) [9,10]. Precipitation is the primary driver of biological activity in arid and semiarid areas, and it further regulates the dynamics of R_s_ by affecting the synergistic and decoupling effect of T_s_ and VWC [5,11]. It has been reported that when the VWC is sufficient, increasing T_s_ will improve R_s_ by increasing the net photosynthesis rate [12], substrate concentration [13], and microbial biomass [14]. However, when water stress occurs, the VWC may be decoupled from the T_s_ and become the primary factor affecting R_s_ [11]. This indicates that there is a threshold of the R_s_ response to T_s_ and VWC and that a change in this threshold may suddenly change the function of the C cycle in the world’s major terrestrial ecosystems [15,16]. However, the response pattern of this threshold to precipitation variation is not clear. Hence, how the responses of the decoupling or synergistic effects between VWC and T_s_ to precipitation variation affect R_s_, particularly in arid and semiarid regions, needs to be addressed.

Temperature sensitivity (Q_10_) is a critical metric for characterizing the relationship between T_s_ and R_s_ [17,18,19], and may be influenced by other biotic and abiotic factors, such as precipitation [18,20], VWC [13], root biomass [17], and microorganisms [19]; Q_10_ can also lead to hysteresis (between T_s_ and R_s_) at multiple scales [21]. Previous studies have shown that the VWC may indirectly affect Q_10_ because the diffusion of extracellular enzymes produced and the available substrates must occur in the liquid phase [13,22]. Moreover, because the thermal conductivity in the liquid phase is higher than that in the solid phase, the elevated VWC caused by precipitation promotes the C transport rate of plants, which further affects the hysteresis relationship [23,24]. However, due to the spatiotemporal variability in R_s_ and the differences among ecosystems, there is still no consensus on how Q_10_ and the hysteresis relationship respond to precipitation variation. Clarifying the responses of Q_10_ and the hysteresis relationship to precipitation variation is crucial for understanding the relationship between soil C pool dynamics and climate change in arid and semiarid areas.

The Loess Plateau (LP) of China is one of the most severely eroded arid and semiarid regions in the world and plays a vital role in global C cycle and climate change research [25]. The LP is also one of the most important winter wheat (*Triticum aestivum* L.)-producing regions in China, and its winter wheat planting area accounts for 44% of the cultivated land area on the LP [26]. Conventional wheat cultivation practices on the LP are relatively vulnerable to climate change, increase the unstable carbon fraction, and lower soil organic C stocks, which in turn aggravate the global C cycle [27]. Moreover, the projections have shown that extreme precipitation events on the LP may increase in the future, which is one of the uncertainty factors in measuring C flux emissions [5]. To date, studies on R_s_ of winter wheat have actively focused on natural precipitation and irrigation [27,28,29]. In contrast, the response of R_s_ to natural precipitation variation has been of less concern, especially the lack of understanding of the threshold change patterns of coupling and decoupling effects among the factors affecting R_s_, which is not conducive to our predictions on the C cycle of farmland systems in the context of climate change. It is urgently necessary to investigate the response of R_s_ to precipitation variation in winter wheat farmland systems through high-frequency R_s_ measurements combined with multi-stage precipitation experiments.

To evaluate the effects of precipitation variation on the R_s_ of winter wheat farmland ecosystems on the Loess Plateau, R_s_ was continuously measured with an automatic soil CO_2_ flux system during the whole growth period from 2019 to 2020, and field simulation experiments with three precipitation levels (P_−30_ = 30% decreased precipitation, CK = ambient precipitation, P_+30_ = 30% increased precipitation) were carried out with a rainfall shelter. The purpose of this study was (1) to quantify the responses of seasonal R_s_ to precipitation variation; (2) to clarify the responses of seasonal R_s_ to T_s_ and VWC under precipitation variation; and (3) to explore the responses of seasonal lag times and Q_10_ to precipitation variation. We hypothesized that (1) precipitation variation would cause significant differences in the VWC, which would significantly affect the seasonal dynamics of the R_s_ and lag time and (2) the synergistic or decoupling response of seasonal R_s_ and Q_10_ to T_s_ and VWC would be dominated by precipitation.

## 2. Materials and Methods

### 2.1. Experimental site Description and Design

The study was conducted from 2019 to 2020 at the Loess Plateau Research Station of Lanzhou University, located in the township of Shishe, Qingyang city, Gansu, China (35°40 N, 107°51 E, with an altitude of 1297 m). This site has a typical semiarid continental monsoon climate, and rainwater is the only water source for crop growth. The mean annual precipitation and temperature are 541 mm and 9.2 °C (1961–2019), respectively, and more than 60% of the precipitation is concentrated in the summer fallow period of winter wheat (from July to September). The average annual pan evaporation is 1504 mm, and the average annual sunshine duration is 2415 h. The soil at the site is silty loam. The average soil bulk density is 1.3 g cm^−3^, the pH is 8.4, the soil organic carbon content is 9.3 g kg^−1^, and the total N content is 0.64 g kg^−1^ in the 0–20 cm soil layer.

The experimental field for the study of the continuous wheat crop under precipitation variation started in September 2018. We used a randomized block design with four replications and three precipitation treatments including 30% decreased precipitation (P_−30_), ambient control (CK), and 30% increased precipitation (P_+30_) treatments. The variation in precipitation in the experimental plots was achieved using rainfall shelters [30], which trapped 30% of the precipitation (P_−30_). The trapped rainfall was channeled to the P_+30_ sites. Each plot was 3 m × 4 m, with a 1 m spacing between plots. We inserted stainless-steel sheets into the ground (40 cm in depth, 10 cm above the ground surface) to prevent lateral water movement. All measurements were conducted in the central area of each plot (2 m × 3 m) to avoid edge effects. In this study, the winter wheat cultivar Longyu 4 was planted in 20 rows of 13.5 g seeds that were sown at a 15 cm spacing on September 25 in 2019 and hand-harvested on June 25 in 2020. Fertilization consisted of 225 kg hm^−2^ of triple superphosphate and 150 kg hm^−2^ of urea (36% was applied as base fertilizer before sowing the winter wheat, and the remaining 64% was applied at the jointing stage).

### 2.2. Measurement of R_s_, T_s_, and VWC

One permanent polyvinyl chloride (PVC) collar (20.3 cm inner diameter, 10 cm height) was installed 6 cm into the soil in the central area of each plot in July 2019. In October 2019, the hourly soil respiration rate of winter wheat in the whole growth period was continuously measured by using an automatic soil carbon dioxide flux system (model LI-8100A fitted with an LI-8150 multiplexer, LI-COR, Lincoln, NE, USA) with LI-104 long-term measurement opaque chambers. The measurement time for each chamber was 3 min and 15 s, comprising a 30 s pre-purge, a 120 s observation period (including a 20 s dead band), and a 45 s post-purge. Any plant re-growth within the measurement collar was manually removed. The hourly T_s_ and VWC at 10 cm depth were measured simultaneously with the soil respiration rates using the 8150-203 soil temperature probe and GS1 soil moisture sensor (LI-COR, Lincoln, NE, USA), respectively. Meteorological data (air temperature and precipitation) were recorded half-hourly using a PC200W automatic meteorological station (Campbell Scientific) placed within 50 m of the experimental field.

### 2.3. Measurements of Soil Profile Moisture and Net Photosynthetic Rate (P_n_)

The soil gravimetric water content in the 0–300 cm (each gradient 20 cm) soil profile of the soil column was determined by the oven drying method at winter wheat sowing and at harvesting. The net photosynthetic rates (P_n_) of winter wheat flag leaves were measured every seven days from anthesis with a portable open gas exchange system (Li-6800-01A, Li-Cor Biosciences Inc., Lincoln, NE, USA) at 9:30–11:00 a.m. on sunny days. The leaves of three plants were measured in each plot.

### 2.4. The Dependence of R_s_ on T_s_ and VWC

The R_s_ was fitted to T_s_ and VWC with empirical exponential and quadratic functions [10,13], respectively:(1)$$R_{s}=a\times e^{b\times T_{s}}$$
(2)$$R_{s}=a\times {\mathrm{VWC}}^{2}+b\times \mathrm{VWC}+c$$
where R_s_, VWC, and T_s_ represent the soil respiration, soil volume water content, and soil temperature, respectively, and a, b, and c are fitting coefficients. Then, the following nonlinear models were used to express the relationships between VWC and T_s_ and R_s_ [31,32]:(3)$$R_{s}=T_{s}\times \mathrm{VWC}/(a\times T_{s}+b\times \mathrm{VWC}+c)$$
(4)$$R_{s}=a\times {T_{s}}^{2}+b\times {\mathrm{VWC}}^{2}+c$$
(5)$$R_{s}=a\times T_{s}+b\times \mathrm{VWC}+c$$
(6)$$R_{s}=a\times {T_{s}}^{b}\times {\mathrm{VWC}}^{c}$$
(7)$$R_{s}=a+b\times (T_{s}\times \mathrm{VWC})$$

The Q_10_ of R_s_ based on Equation (1) was calculated as:(8)$$Q_{10}=e^{10\times b}$$

### 2.5. Statistical Analysis

Due to instrument failure of the LI-8100A during the measurement period (from 5 to 19 October 1 to 11 December 2019 and 1 to 6 March 2020), 12% of CO_2_ flux data were missing. We calculated the cumulative CO_2_ emissions using Matlab’s trapz function with hourly data during the winter wheat whole growth period. The missing data were replaced by the hourly *T_s_* and VWC fitted values with Equation (3) mentioned above. The monthly Q_10_ was analyzed using the short-term Q_10_ derived from fitting the Q_10_ model to synchronized data from a three-day moving window with a one-day step. One-way analysis of variance (ANOVA) followed by Duncan’s post hoc tests was used to perform multiple comparisons of the effect of precipitation on the monthly R_s_ and Q_10_. A linear model (*y = ax + b*) was used to determine the relationship between VWC and the monthly average diurnal dynamic lag time between R_s_ and T_s_. A level of *p* < 0.05 was accepted as significant. To succinctly describe the monthly lag time between the diurnal average T_s_ and R_s_ during the whole growth period of winter wheat in the three precipitation treatments, we chose November 2019 and January, April, and June 2020 to represent the whole growth period, in intervals of 1–2 months, due to the fact that these months contain the key phenological periods of winter wheat, such as seedling, overwintering, jointing, booting, and harvest. All analyses were processed with a combination of MATLAB ver. R2019b (Mathworks Inc., Natick, MA, USA) and SPSS 25 (IBM Corp., Armonk, NY, USA).

## 3. Results

### 3.1. Environmental Conditions and R_s_

The soil temperature (T_s_) exhibited the same temporal pattern as air temperature, and the mean air temperature was 7.43 °C during the winter wheat whole growth period (Figure 1a,b and Figure 2b). The highest and lowest monthly mean T_s_ values occurred in Jun (P_−30_ treatment, 18.64 °C) and Jan 2020 (CK treatment, −0.62 °C), respectively. The T_s_ in the different precipitation treatments showed very similar seasonal variations. The average T_s_ was highest in P_+30_ treatment (8.19 °C), which was 13.9% and 7.2% higher than those in the CK and P_−30_ treatments (*p* < 0.05), respectively (Figure 2b). The variation in VWC was controlled mostly by precipitation events, and the total precipitation was 214.3 mm during the winter wheat whole growth period (Figure 1a,c). There were significant differences in the average VWC throughout the growth period, and the highest VWC, observed in the P_+30_ treatment, was 20.50%, which was 16.6% and 32.5% higher than those in the CK and P_−__3__0_ treatments (*p* < 0.05), respectively (Figure 2a).

Soil respiration (R_s_) exhibited the same temporal pattern as T_s_ and was also affected by the rainfall pulse (Figure 1a,b,d). The highest R_s_ (from 0.38 to 0.41 μmol m^−2^ s^−1^) and the lowest R_s_ (from 2.51 to 2.68 μmol m^−2^ s^−1^) in the three treatments during the winter wheat whole growth period occurred in May and Jan 2020, respectively (Figure 2c). The average R_s_ of the P_−30_ treatment (1.38 μmol m^−2^ s^−^^1^) was significantly higher than those of the other treatments during the whole growth period and was 6.5% and 3.8% higher than those of the CK (1.30 μmol m^−2^ s^−1^) and P_+30_ (1.32 μmol m^−2^ s^−1^) treatments (*p* < 0.05), respectively. The cumulative CO_2_ emission is similar to the monthly average dynamic of R_s_, and the cumulative CO_2_ emissions under P_−30_, CK, and P_+30_ treatments during the winter wheat whole growth period were 406.37, 372.58, and 383.59 g C m^−2^, respectively (Table 1).

### 3.2. Relationships between R_s_ and T_s_, VWC

The exponential model described well the relationship between R_s_ and T_s_ (Figure 3a,c,e). T_s_ explained 85–93% of the seasonal variation in R_s_ (*p* < 0.01), and the R^2^ values increased with an increase in precipitation. Meanwhile, a quadratic model fit the relationship between soil R_s_ and VWC well in the CK and P_+30_ treatments (Figure 3d,f); VWC explained 49–51% of the seasonal variation in R_s_ (*p* < 0.05), and the response of R_s_ to VWC first increased and then decreased. However, the piecewise linear function described well the relationship between R_s_ and VWC when 15% VWC was used as the boundary in the P_−__3__0_ treatment (Figure 3b), and VWC explained 40% and 56% of the seasonal variation in R_s_, respectively (*p* < 0.01). The estimated threshold of optimal soil moisture for R_s_ was in the range of 15.00–17.07% in the three treatments during the winter wheat whole growth period.

The application of the interactive functions for T_s_ and VWC explained 72–93% of the variation in seasonal R_s_ in the three precipitation treatments (*p* < 0.01, Table 2, Equations (3)–(7)). Equations (3), (4), and (6) exhibited a better representation (R^2^ = 85–93%) of the relationship than the single-factor functions (R^2^ = 84–88%) using either seasonal T_s_ or VWC in the P_−30_ and CK treatments (Figure 3 and Table 2). However, the inclusion of the VWC functions in Equations (3)–(7) did not improve the determination coefficients for seasonal R_s_ (R^2^ = 72–90%) compared with the single-factor functions using T_s_ (R^2^ = 92%).

### 3.3. Hysteresis between R_s_ and T_s_

There were obvious phase differences in the diurnal dynamics of the monthly average T_s_ and R_s_ of winter wheat in the three precipitation treatments. R_s_ consistently peaked earlier than T_s_ (Figure 4), as shown by the lag in the R_s_-T_s_ relationship. In the winter wheat whole growth period, the monthly lag time increased at first and then decreased (Figure 5); it was lowest in Oct 2019 (at 1 h, 2 h, and 2 h in the P_−__3__0_, CK, and P_+30_ treatments, respectively) and highest in Jan 2020 (8 h in three treatments).

The diurnal dynamics of the monthly average T_s_ and R_s_ showed an elliptical trajectory with a rotated clockwise direction in the three treatments (Figure 5). The peak temperature difference in R_s_ and T_s_ was lowest in the P_−30_ treatment, which was significantly lower than those in the other treatments (*p* < 0.05). The seasonal lag time (between diel R_s_ and T_s_) was negatively and linearly correlated with VWC in the three treatments during the winter wheat whole growth period (*p* < 0.01, Figure 6).

### 3.4. Temperature Sensitivity (Q_10_) of R_s_

The ranges of monthly Q_10_ under the P_−30_, CK, and P_+30_ treatments were 0.75–3.25, 1.05–2.08, and 0.99–3.11 during the whole growth period, respectively (Figure 7). The seasonal difference in Q_10_ (from 1.84 to 2.14) was not significant (*p* > 0.05, Figure 7). The monthly Q_10_ decreased at first and then increased, and the lowest values (in May 2020) were 0.75, 1.05, and 0.99 in P_−30_, CK, and P_+30_ treatments, respectively.

The seasonal Q_10_ exhibited a negative polynomial correlation with the seasonal average T_s_ in the P_−30_ treatment, and Q_10_ initially increased and then decreased with T_s_ (Figure 8a). However, the seasonal Q_10_ increased with the increase in T_s_ in the CK and P_+30_ treatments (Figure 8c,e), as did the relationship between the seasonal Q_10_ and the VWC in the three precipitation treatments (Figure 8b,d,f).

## 4. Discussion

### 4.1. Effects of Precipitation on R_s_ in Winter Wheat Systems

Changes in both the distribution and amount of precipitation have significant effects on soil CO_2_ emissions by regulating VWC; this is especially true in rain-fed agricultural areas, where precipitation is the primary driver of biological activity [26]. Generally, an increase in VWC will increase the aboveground and underground biomass, as well as substrate concentrations of plants [5,13], thereby enhancing R_s_ [14]. However, a previous study also found that a decrease in VWC increased T_s_ and then increased R_s_ [33]. In our study, the average R_s_ of the P_−30_ treatment was significantly higher than those of the CK and P_+30_ treatments during the whole growth period (Figure 2c, *p* < 0.05). This was mainly due to the variation between T_s_ and VWC in the different seasons and eventually led to the seasonal average R_s_ of the P_−30_ treatment being higher than those of other treatments [34]. For example, from January to March 2020, the VWC of the P_−30_ treatment was significantly lower than those of the other treatments and induced a significant increase in T_s_ (Figure 2a,b), while higher winter temperatures may promote microbial activity, resulting in a significant increase in R_s_ [35]. Therefore, the variation in the response of VWC and T_s_ to the precipitation in different seasons may be the main reasons for the differing responses of R_s_ to VWC [34].

Similar to the pattern of seasonal average R_s_, the cumulative CO_2_ emissions during the winter wheat whole growth period were 406.37, 372.58, and 383.59 g C m^−2^ under the three precipitation treatments, respectively (Table 1), which is consistent with the study of irrigation gradient in Northwest China (cumulative CO_2_ emissions ranged from 335 to 448 g C m^−2^ season^−1^) [36], but lower than the study in the North China Plain (cumulative CO_2_ emissions ranged from 548.0 to 979.2 g C m^−2^ season^−1^) [37]. These differences may be due to the higher average annual temperature and rainfall in the North China Plain than in the Loess Plateau.

### 4.2. Responses of R_s_ to T_s_ and VWC Coupling to Precipitation Variation

Soil temperature (T_s_) is the most important environmental factor controlling the seasonal variation in soil respiration [38,39]. Higher T_s_ can increase the production of root exudates; this accelerates the decomposition rate of the substrate by microorganisms, which further increases R_s_ [14]. Our results showing that R_s_ had an extremely significant exponential correlation with T_s_ in the three precipitation treatments (*p* < 0.01, Figure 3a,c,e) and that the R^2^ values increased with precipitation (from 85% to 93%) are in line with previous studies [35,40]. In addition to T_s,_ the VWC is an important environmental factor affecting seasonal soil CO_2_ emissions [14]. In rain-fed agricultural areas, precipitation is the primary driver changing the VWC and regulating biological activity [26,40]. Many studies have shown that the response of R_s_ to VWC increases at first and then decreases and that there is a VWC threshold [29,41]. This is mainly due to a low VWC triggering cell dehydration and soil microbial death; this reduces microbial biomass and plant biomass, causing lower substrate concentrations and weakened organic matter mineralization and, therefore, a decrease in R_s_ [5,14]. If this threshold is exceeded, increasing soil moisture may cause soil pore saturation, increase the leaching of soluble matter, and inhibit the activities of microorganisms and roots, thus inhibiting R_s_ [41,42]. In this study, there was a significant correlation between the VWC and R_s_, and R_s_ increased at first and then decreased with the VWC in the CK and P_+30_ treatments (Figure 3d,f, *p* < 0.05). The threshold of the optimal VWC for R_s_ was approximately 15.00–17.07% in the three treatments, which is consistent with the results of Tan [41]. However, the piecewise linear function described well the relationship between R_s_ and the VWC when 15% VWC was used as the boundary in the P_−__3__0_ treatment (Figure 3b), and the response of R_s_ to the VWC showed a bimodal trend with a mean threshold above 17.07%. The coupling of the legacy and priming effects of precipitation can explain this phenomenon. R_s_ mainly occurs in surface soils [43], and the lower the VWC of the soil layer, the higher the threshold of precipitation that triggers the R_s_ response [30]. After one year of the precipitation variation treatment (Oct 2019) (Figure 3a), the soil moisture profile at 0–100 cm in the soil P_−__3__0_ treatment was significantly lower than those in the CK and P_+30_ treatments (Appendix A); this resulted in the accumulation of substrates such as dissolved organic carbon (DOC) [44]. The physical replacement of CO_2_ from soil pores after the precipitation event may have contributed to the higher CO_2_ efflux under drier conditions, while soil rewetting would have promoted the dissolution of the substrate, accelerated root growth due to microbial metabolism activity [5,44], and exacerbated the effect of water restriction on CO_2_ emissions under drought, resulting in a more sensitive response of R_s_ to precipitation under the P_−__3__0_ treatment [14,45].

Compared with the influence of a single factor, there was a stronger synergistic effect of the relationship between T_s_ and VWC on R_s_. On the one hand, changes in soil water content have a significant impact on T_s_ [33]; on the other hand, the soil water content can enhance the effect of R_s_ on T_s_ [16]. Therefore, a bivariate model (that includes both T_s_ and VWC) can be used to evaluate the changing trend of R_s_ more accurately [16]. In this study, the application of interactive functions of T_s_ and VWC (Equations (3), (4) and (6)) explained 85–93% of the variation in soil respiration in the P_−30_ and CK treatments (Figure 3 and Table 2), and these functions had higher R^2^ values than the single-factor functions (R^2^ = 84–88%). However, the inclusion of the VWC function in Equations (3)–(7) did not improve the determination coefficients of R_s_ (R^2^ = 72–90%) compared with those of the single-factor functions using T_s_ (R^2^ = 92%) in the P_+30_ treatment. This may be attributed to the decoupling of soil moisture from T_s_ under high-moisture (not water-limited) conditions [5,11], supports our second hypothesis.

### 4.3. The Response of Diel Hysteresis to Precipitation

The daily hysteresis between R_s_ and T_s_ is one of the uncertainty factors in soil carbon flux simulation models and has received increasing attention [46]. Across the diurnal cycles, our results show a significant hysteresis between the hourly R_s_ and T_s_ at 10 cm depth, with R_s_ peaking earlier than T_s_ (Figure 4). Similar hysteresis relationships between diurnal R_s_ and T_s_ have also been observed in other ecosystems [47,48]. Conversely, studies on an oak-grass savanna [12], mixed conifer and oak forest [49], and wheat fields [50] reported that T_s_ peaked earlier than R_s_; this discrepancy may be due to the variation in the hysteresis relationship between T_s_ and R_s_ caused by the legacy effects of different ecological hydrothermal relationships [51]. Furthermore, in this study, R_s_ exhibited diurnal clockwise hysteresis loops with T_s_ in the three precipitation treatments (Figure 5). A similar hysteresis loop phenomenon has also been observed in other studies [46,52].

Both biological and physical processes may contribute to the observed diurnal hysteresis. Our study found that the seasonal lag time between the diurnal R_s_ and T_s_ was negatively correlated with the VWC in the three precipitation treatments (Figure 6), supporting our first hypothesis. A similar relationship between VWC and lag time has also been observed in desert ecosystems in northwestern China [46,53]. This may be attributed to the heat transfer rate of wet soil being faster than that of dry soil [24]; the higher water content of surface soil increases the heat transfer rate of soil and shortens the lag time between autotrophic respiration and heterotrophic respiration in the P_+30_ treatment. Meanwhile, increasing precipitation significantly increased the stomatal conductance, transpiration rate, and intercellular CO_2_ concentration of the winter wheat leaves, thus significantly increasing the photosynthetic rate (Appendix A) and metabolic rate [23]; this shortened the lag time caused by photosynthetic carbon transport. This may be an explanation for the observed dynamics of seasonal lag time. From November 2019 to February 2020, no precipitation events greater than 5 mm occurred, and evapotranspiration resulted in a gradual decrease in surface VWC (Figure 1a,c). After March 2020, the gradual increase in precipitation events led to an increase in surface VWC, so the diurnal seasonality lag time first increased and then decreased (Figure 5).

### 4.4. The Response of Seasonal Q_10_ to Precipitation

Q_10_ not only reflects temperature sensitivity, but also integrates the responses of root biomass, litter input, water conditions, and unknown variables [19]. A small error in Q_10_ may lead to large inaccuracies in carbon dynamic estimations [54]. Generally, T_s_ and VWC are the most important abiotic factors affecting Q_10_. In our study, there was a positive correlation between seasonal Q_10_ and T_s_ in the CK and P_+30_ treatments (Figure 8c,e). However, the seasonal Q_10_ exhibited a negative polynomial correlation with seasonal average T_s_ in the P_−30_ treatment (Figure 8a). This may be attributed to the VWC being higher under CK and P_+30_ than under P_−30_ treatments (Figure 2a); the increased T_s_ would have favored the diffusion of the soluble substrate, which would have increased Q_10_ [13]. In contrast, the relatively low soil moisture environment decoupled VWC from T_s_, and Q_10_ may be limited by VWC under drought stress, which supports hypothesis 2. An increase in T_s_ would have accelerated the evaporation of VWC in the surface layer when VWC exceeded the threshold under the P_−30_ treatment, which would have limited the utilization of soluble substrate; therefore, the response of Q_10_ to T_s_ tended to increase at first and then decrease [55].

The VWC is also an important environmental factor affecting Q_10_. Some studies have shown that Q_10_ has a negative quadratic relationship with the VWC and that the response of Q_10_ to the VWC increases at first and then decreases [13,17]. This phenomenon may occur due to the following reasons: first, the lower VWC may limit the supply of respiratory substrates and thus reduce Q_10_ [13]. Second, higher soil moisture can also reduce Q_10_ by limiting the diffusion rate of O_2_; the diffusion rate of O_2_ through water is much slower than that through air, and the decomposition activity of aerobic microorganisms is therefore inhibited due to hypoxia [13,42]. Unlike previous studies, we found a positive correlation between seasonal Q_10_ and the VWC (Figure 8b,d,f). The discrepancies from previous studies may be attributed to the fact that the VWC in this study did not reach the threshold of the Q_10_ slave response to the VWC. A similar response pattern has been detected in another study [18].

## 5. Conclusions

This manipulation experiment investigated the effect of precipitation variation on the temporal variation in R_s_ by recording high-frequency data in a winter wheat farmland system on the semiarid Loess Plateau of China. This study found that the response of seasonal R_s_ to precipitation variation was affected by the synergistic influence of the precipitation legacy and priming effects; hence, reducing rainfall significantly increased the average R_s_. The cumulative CO_2_ emissions under P_−30_, CK, and P_+30_ treatments during the winter wheat whole growth period were 406.37, 372.58, and 383.59 g C m^−2^, respectively. The synergistic effects of T_s_ and VWC best explained the seasonal variations in R_s_ and Q_10_. However, the increase and decrease in precipitation led to the decoupling of R_s_ and Q_10_ responses to T_s_ and VWC, respectively. The seasonal dynamics of the diurnal lag time were significantly negatively correlated with the VWC, and the decrease in precipitation increased the threshold of the R_s_ response to the VWC. Clarifying the synergistic and decoupling response of R_s_ and Q_10_ to T_s_ and the VWC and the threshold change of R_s_ to the VWC under precipitation variation scenarios can benefit the prediction of the future C balances in agroecosystems in semiarid regions under climate change. To reduce the possible limitations of short-term studies, future long-term precipitation simulation studies are needed to further clarify the relationship between soil carbon emissions and precipitation variation.

## Figures and Tables

**Figure 1 ijerph-19-06915-f001:**
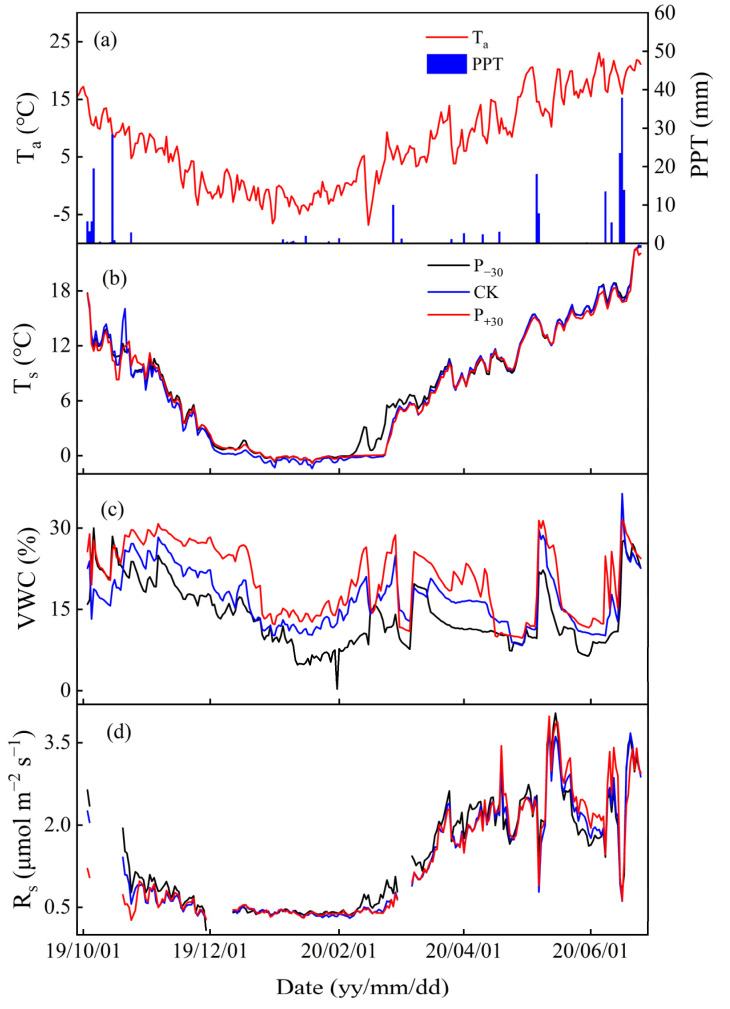
Variations in air temperature, T_a_ (**a**); ambient precipitation, PPT (**a**); soil temperature, T_s_ (**b**); soil volumetric water content, VWC (**c**); and soil respiration, R_s_ (**d**) in the three precipitation treatments (P_−30_ = 30% decreased precipitation, CK = natural precipitation, P_+30_ = 30% increased precipitation) during the winter wheat whole growth period. T_s_ and VWC were measured at 10 cm depth.

**Figure 2 ijerph-19-06915-f002:**
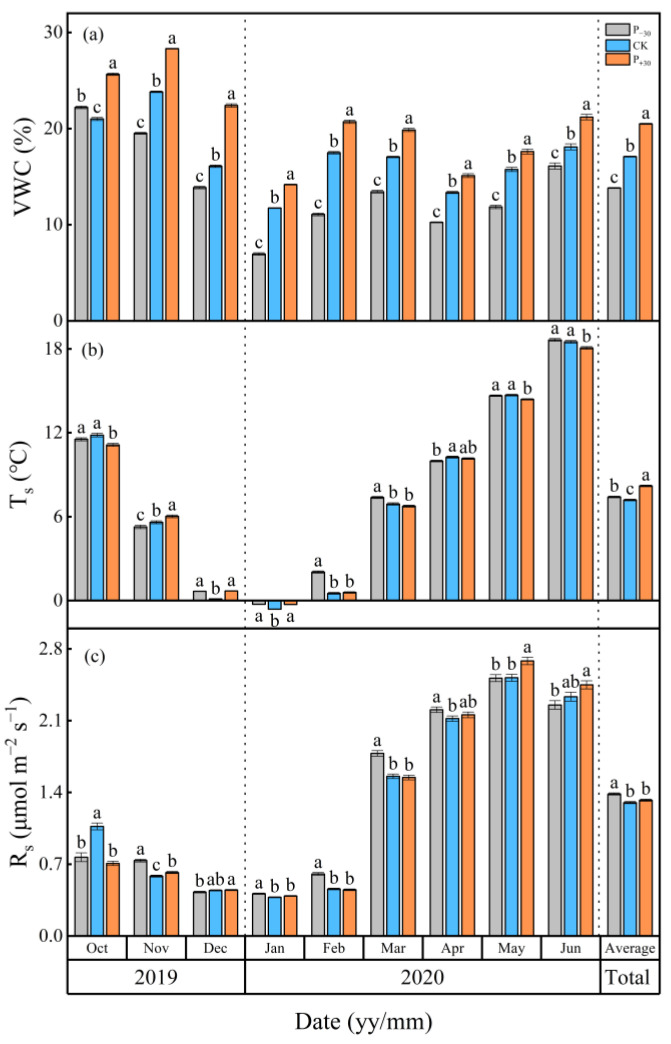
Monthly average dynamics of soil volumetric water content, VWC (**a**); soil temperature, T_s_ (**b**); and soil respiration (R_s_) (**c**) in three precipitation treatments (P_−30_ = 30% decreased precipitation, CK = ambient precipitation, P_+30_ = 30% increased precipitation) during the winter wheat whole growth period. T_s_ and VWC were measured at 10 cm depth. Vertical bars represent standard errors of the mean. Different letters represent significant differences (*p* < 0.05) among the three precipitation treatments in the same month. Three precipitation treatments containing the same letter in the same month indicate nonsignificant differences (*p* > 0.05).

**Figure 3 ijerph-19-06915-f003:**
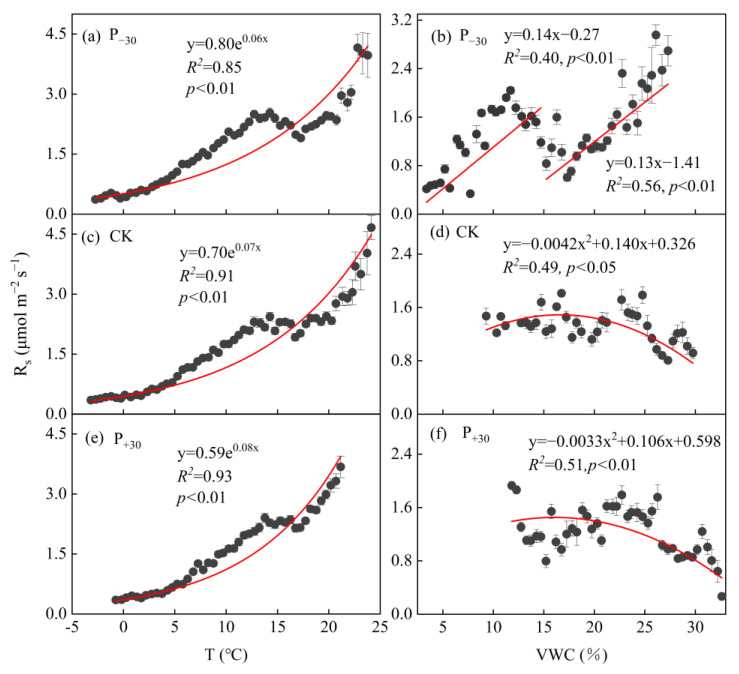
Correlations of soil respiration (R_s_) with soil temperature (T_s_) (**a**,**c**,**e**) and soil volumetric water content (VWC) (**b**,**d**,**f**) in the different precipitation treatments (P_−30_ = 30% decreased precipitation, CK = ambient precipitation, P_+30_ = 30% increased precipitation) during the winter wheat whole growth period. The exponential function was used for fitting R_s_ and T_s_ (R_s_ = a×e^bT^). The piecewise linear function (R_s_ = a × VWC + b) of the response of R_s_ to VWC in the P_−30_ treatment (bounded by 15% of VWC) and the logarithmic function for R_s_ and VWC (R_s_ = a × VWC ^2^+ b × VWC + c) in the CK and P_+30_ treatments, respectively. The red lines indicates the significant relationship between R_s_ and T_s_ or VWC. Error bars represent standard errors of the mean. Hourly values were bin-averaged every 0.5 intervals of T_s_ and VWC.

**Figure 4 ijerph-19-06915-f004:**
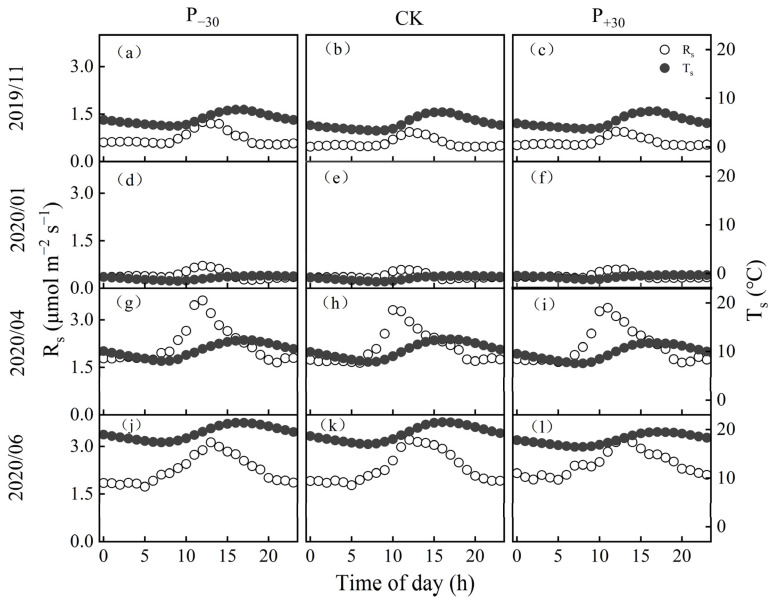
Mean monthly diel cycles of soil respiration (R_s_, open points) and soil temperature (T_s_, solid points) at a depth of 10 cm in November 2019 (**a**–**c**) and January (**d**–**f**), April (**g**–**i**), and June (**j**–**l**) 2020.

**Figure 5 ijerph-19-06915-f005:**
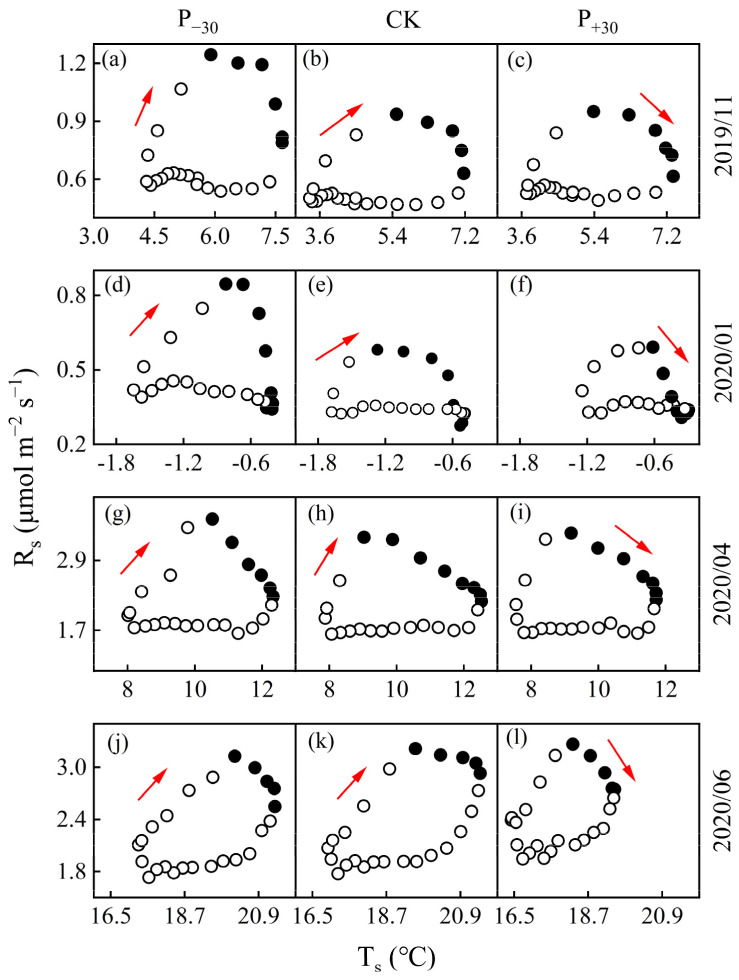
Mean monthly diel cycles of soil respiration (R_s_) and soil temperature (T_s_) at a depth of 10 cm in November 2019 (**a**–**c**) and January (**d**–**f**), April (**g**–**i**), and June (**j**–**l**) 2020. The number of solid points and the arrow indicate the monthly average diurnal dynamic lag time between R_s_ and T_s_ and the direction of the diel cycle, respectively.

**Figure 6 ijerph-19-06915-f006:**
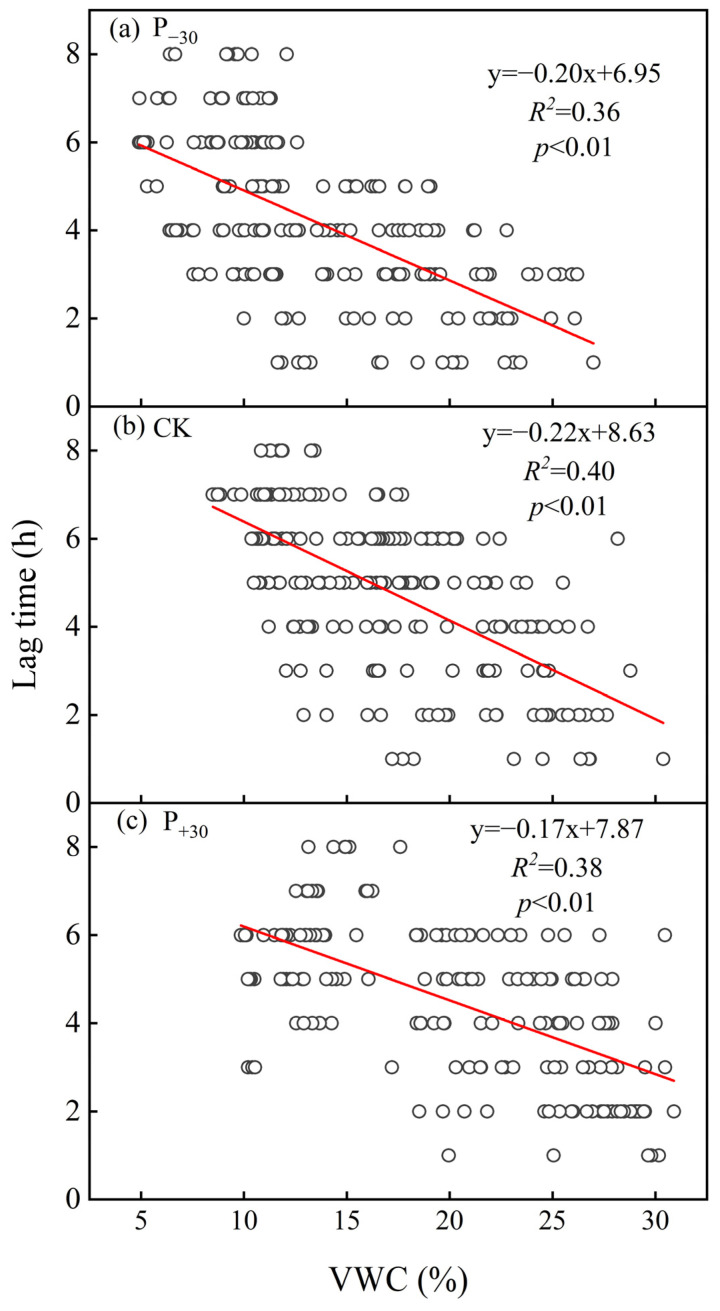
Relationship between soil volumetric water content (VWC) and lag time in three precipitation treatments (P_−30_ (**a**), CK (**b**) and P_+30_ (**c**)) and VWC at 10 cm soil depth. The lag times were calculated by a cross-correlation analysis using a three-day moving window with a one-day step. The solid line is fitted using linear regression.

**Figure 7 ijerph-19-06915-f007:**
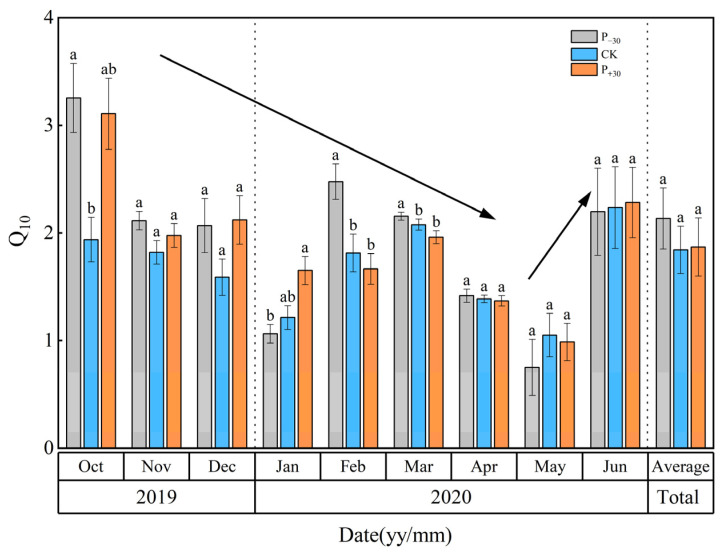
Monthly average dynamics of temperature sensitivity (Q_10_) in three precipitation treatments during the winter wheat whole growth period. Vertical bars represent standard errors of the mean. Different letters represent significant differences (*p* < 0.05) among the three precipitation treatments in the same month. Three precipitation treatments containing the same letter in the same month indicate nonsignificant differences (*p* > 0.05).

**Figure 8 ijerph-19-06915-f008:**
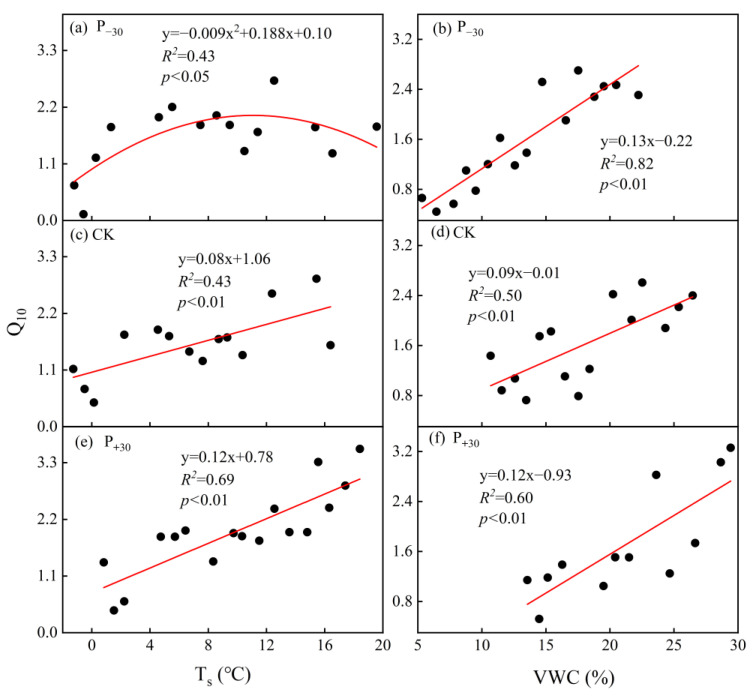
Relationships between seasonal soil temperature (T_s_), soil volumetric water content (VWC), and Q_10_ in the different precipitation treatments during the winter wheat whole growth period. The red lines indicates the significant relationship between Q_10_ and T_s_ (**a**,**c**,**e**) or VWC (**b**,**d**,**f**).

**Table 1 ijerph-19-06915-t001:** Monthly and total cumulative CO_2_ emissions in three precipitation treatments (P_−30_ = 30% decreased precipitation, CK = natural precipitation, P_+30_ = 30% increased precipitation) during the winter wheat whole growth period.

Month	Treatment	Cumulative CO_2_ Emission (g C m^−2^)	Month	Treatment	Cumulative CO_2_ Emission (g C m^−2^)
October 2019	P_−30_	65.23	March 2020	P_−30_	59.12
	CK	50.42		CK	48.45
	P_+30_	50.38		P_+30_	49.44
November 2019	P_−30_	21.53	April 2020	P_−30_	68.54
	CK	17.59		CK	65.87
	P_+30_	18.77		P_+30_	66.99
December 2019	P_−30_	13.96	May 2020	P_−30_	82.47
	CK	13.99		CK	82.07
	P_+30_	14.53		P_+30_	87.33
January 2020	P_−30_	13.18	June 2020	P_−30_	58.33
	CK	12.16		CK	60.44
	P_+30_	12.54		P_+30_	63.41
February 2020	P_−30_	18.53	Total	P_−30_	406.37
	CK	14.12		CK	372.58
	P_+30_	13.72		P_+30_	383.59

“Total” is the cumulative CO_2_ emission of winter wheat during the whole growth period.

**Table 2 ijerph-19-06915-t002:** Regression equations of soil respiration (R_s_) against soil temperature (T_s_) and soil volumetric water content (VWC).

No.	Model	P	n	df	a	b	c	*R^2^*	*p*
3	R_s_ = T_s_ × VWC/(a × T_s_ + b × VWC + c)	P_−30_	47	44	−8.43	8.57	71.91	0.88	<0.01
		CK	44	41	−6.02	7.38	57.41	0.92	<0.01
		P_+30_	47	44	−8.07	5.56	112.65	0.88	<0.01
4	R_s_ = a × T_s_^2^ + b × VWC^2^ + c	P_−30_	47	44	0.01	−0.08	0.37	0.86	<0.01
		CK	44	41	0.01	−0.06	0.36	0.93	<0.01
		P_+30_	47	44	0.01	0.08	−0.24	0.89	<0.01
5	R_s_ = a × T_s_ + b × VWC + c	P_−30_	47	44	0.22	0.02	−0.53	0.79	<0.01
		CK	44	41	0.22	−0.01	−0.21	0.85	<0.01
		P_+30_	47	44	0.28	0.01	−1.28	0.83	<0.01
6	R_s_ = a × T_s_^b^ × VWC^c^	P_−30_	47	44	0.03	1.98	−0.27	0.85	<0.01
		CK	44	41	0.03	1.8	−0.05	0.92	<0.01
		P_+30_	47	44	0.01	2.16	0.16	0.90	<0.01
7	R_s_ = a + b (T_s_ × VWC)	P_−30_	47	45	0.11	0.01	-	0.76	<0.01
		CK	44	42	−0.26	0.01	-	0.81	<0.01
		P_+30_	47	42	−1.08	0.01	-	0.72	<0.01

The model number is consistent with that mentioned in the materials and methods, n is the number of bins averaged every 0.5 intervals of T_s_ and VWC, and df is the degree of freedom.

## Data Availability

Not applicable.

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
