# Peer review of "Effect of Precipitation Variation on Soil Respiration in Rain-Fed Winter Wheat Systems on the Loess Plateau, China"

_ijerph, 2022, doi:10.3390/ijerph19116915_

Round 1

Reviewer 1 Report

Comments:

1) Abstract – I think the authors should expand all the abbreviations in the abstract (e.g., Ts). Authors should also consider including keywords which are not provided in the title.

2) Introduction – Lines 43-45 could be improved by splitting the sentence into two statements. Also, many of the references cited in the Introduction are not recent (publications published more than five years) – it would be great if the authors could include some recent references.

3) Methodology – Was LI-8100A (Line 132) or LI-8000 (Line 158) used in the study, or both? The LI-8100 was broken during the measurement period, according to the authors (Lines 158-159). To improve data consistency, I recommend that the authors calculate the cumulative CO2 emissions for the entire experiment and remove the methods/data from the LI-8100. Why are there underlined statements between Lines 168 and 171?

4) The findings have been properly discussed. Both Results and Discussion sections, however, feature a serious flaw in which various statements in almost every sections/subsections (beginning Line 174 up to Line 354) were underlined as if the script was the edited rather than the final version.

5) Conclusions – Limitations of the study should be included, and the authors should make clear recommendations for future research.

Author Response

Dear Editor and Reviewers,

Thank you very much for your constructive comments and suggestions on our manuscript entitled “Effect of precipitation variation on soil respiration in rain-fed winter wheat systems on the Loess Plateau, China” (ijerph-1738104). We have revised the manuscript in accordance with the reviewers' comments. The specific revisions are shown in the reply letter below and the newly submitted manuscript with track changes.

We hope that the revised manuscript and our accompanying responses will be sufficient to make our manuscript suitable for publication in International Journal of Environmental Research and Public Health.

Best regards and yours sincerely,

Yuying Shen

Email: yy.shen@lzu.edu.cn

Reviewer 1:

1)Abstract: I think the authors should expand all the abbreviations in the abstract (e.g., Ts). Authors should also consider including keywords which are not provided in the title.

Response:Thanks for your suggestion. We revised as suggested (Page 1 Line 20-21). Meanwhile, we replaced the keyword “winter wheat” with “decouple” (Page 1 Line 31).

2) Introduction – Lines 43-45 could be improved by splitting the sentence into two statements. Also, many of the references cited in the Introduction are not recent (publications published more than five years) – it would be great if the authors could include some recent references.

Response:We appreciate this advice; it has been separated to two sentences (Page 2 Line 45). Also, we added some papers published within the last two years to the introduction.

3) Methodology – Was LI-8100A (Line 132) or LI-8000 (Line 158) used in the study, or both? The LI-8100 was broken during the measurement period, according to the authors (Lines 158-159). To improve data consistency, I recommend that the authors calculate the cumulative CO2 emissions for the entire experiment and remove the methods/data from the LI-8100. Why are there underlined statements between Lines 168 and 171?

Response:Sorry for the mistake. “8100” was replaced with “8100A” (Page 4 Line 159).

We have also added notes to the cumulative CO2 emissions during the whole growth period in Table 1 (Page 8 Line 216-217). I think it is better for the reader to understand the operation of the instrument by keeping the operating parameters of the LI-8100A. Sorry for the underlining in the manuscript layout, we have corrected it.

4) The findings have been properly discussed. Both Results and Discussion sections, however, feature a serious flaw in which various statements in almost every sections/subsections (beginning Line 174 up to Line 354) were underlined as if the script was the edited rather than the final version.

Response:We are sorry for the underlining in the layout of the manuscript, we have corrected it and we confirm that this is the final version of the manuscript.

5) Conclusions – Limitations of the study should be included, and the authors should make clear recommendations for future research.

Response:We appreciate this advice. We added limitations of the study and perspectives for future research to conclusions (Page 17 Line 448-451).

Reviewer 2 Report

This paper investigated the effect of precipitation variation on the temperature variation in Rs in a wheat farm land system on the Loess Plateau  of China. The article has clear logic, solid data and smooth expression. I suggest that the paper can be accepted in the current form,

A few suggestions are listed as follows;

1. Equation 2  can be deleted under the line 155, since it has been mentioned above.

2. In Figure 2 and Figure 7, letters a, b and c represent the significant differences among treatments in the same month, however, the meaning/values of the letters are still not clarified.

Author Response

Dear Editor and Reviewers,

Thank you very much for your constructive comments and suggestions on our manuscript entitled “Effect of precipitation variation on soil respiration in rain-fed winter wheat systems on the Loess Plateau, China” (ijerph-1738104). We have revised the manuscript in accordance with the reviewers' comments. The specific revisions are shown in the reply letter below and the newly submitted manuscript with track changes.

We hope that the revised manuscript and our accompanying responses will be sufficient to make our manuscript suitable for publication in International Journal of Environmental Research and Public Health.

Best regards and yours sincerely,

Yuying Shen

Email: yy.shen@lzu.edu.cn

Reviewer 2:

This paper investigated the effect of precipitation variation on the temperature variation in Rs in a wheat farm land system on the Loess Plateau of China. The article has clear logic, solid data and smooth expression. I suggest that the paper can be accepted in the current form.

Response:Thank you for the support for our study and the suggestions on manuscript. The manuscript has been revised and improved following your suggestions and comments.

1. Equation 2 can be deleted under the line 155, since it has been mentioned above.

Response:We appreciate this comment. We revised as suggested (Page 4 Line 156).

2. In Figure 2 and Figure 7, letters a, b and c represent the significant differences among treatments in the same month, however, the meaning/values of the letters are still not clarified.

Response:We are sorry for the unclear description. We have added to the description of multiple comparative significance (Page 7 Line 201-203 and Page 13 Line 286-289).

Reviewer 3 Report

This manuscript presents the results of an experimental study whose aim was to study the effect of precipitation variation on soil respiration in winter wheat farmland systems. Overall, I found the manuscript to be very well written. The topic is well-motivated, the methods are clear, the results are well-presented and the discussion is well-rounded. I am pleased to recommend this manuscript for publication. I have outlined some minor text edits which are essentially requests for clarifications:

1. Please define the symbols Ts and C in the abstract as well.

2. Section 2.2, please expand the acronym PVC.

3. Section 2.5: The end date of the missing measurement period for October 2019 is missing.

4. Lag time: Please clarify the definition of lag time when it is first introduced in section 2.5.

5. You chose November, January, April and June to represent the whole growth period. It is not clear to me why this choice was made. A sentence or two further justifying this choice could be useful.

6. Section 3.1, line 177: can you clarify which month refers to the winter wheat growth period?

7. Figure 1(a): Does PPT refer to the ambient precipitation (CK)?

8. Figure 2: Can you clarify what does ‘ab’ mean? Does it mean that it is not significantly different from the treatments marked ‘a’ and ‘b’ when checked using the Duncan post hoc test?

9. Section 3.2, lines 238-240: This sentence seems to contradict the previous sentence. Do you mean that inclusion of VWC functions did not improve determination coefficients for the P+30 treatment? Please clarify.

10. Figure 5: I found the use of solid points slightly confusing. If I understand correctly, the location of solid points does not matter, only their number. Some additional clarification could help.

Author Response

Dear Editor and Reviewers,

Thank you very much for your constructive comments and suggestions on our manuscript entitled “Effect of precipitation variation on soil respiration in rain-fed winter wheat systems on the Loess Plateau, China” (ijerph-1738104). We have revised the manuscript in accordance with the reviewers' comments. The specific revisions are shown in the reply letter below and the newly submitted manuscript with track changes.

We hope that the revised manuscript and our accompanying responses will be sufficient to make our manuscript suitable for publication in International Journal of Environmental Research and Public Health.

Best regards and yours sincerely,

Yuying Shen

Email: yy.shen@lzu.edu.cn

Reviewer 3:

This manuscript presents the results of an experimental study whose aim was to study the effect of precipitation variation on soil respiration in winter wheat farmland systems. Overall, I found the manuscript to be very well written. The topic is well-motivated, the methods are clear, the results are well-presented and the discussion is well-rounded. I am pleased to recommend this manuscript for publication.

Response:Thank you for the support for our study and the suggestions on manuscript. The manuscript has been revised and improved following your suggestions and comments.

1. Please define the symbols Ts and C in the abstract as well.

Response:Thanks for your suggestion. We have added full names to the abbreviations (Page 1 Line 9 and 20).

2. Section 2.2, please expand the acronym PVC.

Response:Thanks for your suggestion. We have added full names to the abbreviations (Page 3 Line 130).

3. Section 2.5: The end date of the missing measurement period for October 2019 is missing.

Response:We are sorry for the unclear description. We have improved the presentation (Page 4 Line 160).

4. Lag time: Please clarify the definition of lag time when it is first introduced in section 2.5.

Response:Thanks for your suggestion. We have improved the presentation (Page 4 and 5 Line 168 and 169).

5. You chose November, January, April and June to represent the whole growth period. It is not clear to me why this choice was made. A sentence or two further justifying this choice could be useful.

Response:We are sorry for the unclear description. Due to the fact that these months contain the key phenological periods of winter wheat, such as seedling, overwintering, jointing, booting and harvest. We have made additional notes in the manuscript (Page 5 Line 173-175).

6. Section 3.1, line 177: can you clarify which month refers to the winter wheat growth period?

Response:We are sorry for the unclear description. “Winter wheat growth period” represents the time from sowing to harvest of winter wheat. We have replaced “growth period” with “whole growth period” in the manuscript.

7. Figure 1(a): Does PPT refer to the ambient precipitation (CK)?

Response:We are sorry for the unclear description. We have replaced “precipitation” with “ambient precipitation” in the manuscript (Page 6 Line 192).

8. Figure 2: Can you clarify what does ‘ab’ mean? Does it mean that it is not significantly different from the treatments marked ‘a’ and ‘b’ when checked using the Duncan post hoc test?

Response:We are sorry for the unclear description. We have added to the description of multiple comparative significance (Page 7 Line 201-203 and Page 13 Line 286-289).

9. Section 3.2, lines 238-240: This sentence seems to contradict the previous sentence. Do you mean that inclusion of VWC functions did not improve determination coefficients for the P+30 treatment? Please clarify.

Response:We are sorry for the unclear description. Soil temperature and soil water content synergistically can better explain the variation in soil respiration, but the explanation of the variation in soil respiration may be reduced when they are decoupled, and the results are discussed in section 4.2 of the manuscript.

10. Figure 5: I found the use of solid points slightly confusing. If I understand correctly, the location of solid points does not matter, only their number. Some additional clarification could help.

Response:We are sorry for the unclear description. We express the lag time of the diurnal dynamics of soil temperature and soil respiration in terms of the number of solid points, emphasizing that it is the number of solid points, i.e. the length of the lag time, and we have improved the expression (Page 11 Line 264-265).
